# Changes in infant non-nutritive sucking throughout a suck sample at 3-months of age

**Emily Zimmerman**[1]*, **Thomas Carpenito**[2], **Alaina Martens**[1]

**1** Department of Communication Sciences & Disorders, Northeastern University, Boston, Massachusetts, United States of America, **2** Department of Health Sciences, Northeastern University, Boston, Massachusetts, United States of America

* e.zimmerman@neu.edu

## Abstract

The goal of this study was to compare how infants' non-nutritive suck (NNS) changes throughout a suck sample. Fifty-four full-term infants (57% male) completed this study at, on average, 3.03 (SD .31) months of age. These infants sucked on our custom research pacifier for approximately five minutes. Infants produced, on average, 14.50 suck bursts during the sample. NNS data was pooled across subjects and breakpoint analyses were completed to determine if there were changes in their NNS patterning. Breakpoints were evident for NNS cycles per burst at burst numbers 18 and 34, and for amplitude ($cmH_2O$) at burst numbers 18 and 29. No breakpoints were present for NNS frequency. Infants exhibit changes in their suck physiology across burst number. When assessing suck, developmental specialists should observe more than one suck burst to attain a more valid and appropriate scope of the infant's suck ability.

## Introduction

Non-nutritive suck, or NNS, is a suck pattern characterized by the absence of nutrient delivery [1]. Infant suck begins *in utero* at approximately 15 weeks' gestational age (GA) [1] and is stable and well-patterned by 34 weeks' GA [2]. NNS physiology has a stereotypical burst-pause pattern, with an intra-burst frequency of 2 Hz and each burst containing 6–12 suck cycles [3].

NNS neural circuitry is highly adaptable to descending cortical inputs, as well as to mechanosensory inputs from the periphery. Because of this specialized circuitry, NNS can be modified by sensory inputs, such as tactile and visual stimulation [4, 5]. The NNS signal can also be altered if infants have different sensory experiences or sensory deprivations. Infants born prematurely have reduced NNS patterning [6], as do infants who experience comorbidities, such as respiratory distress syndrome or small for gestational age [7, 8].

Infant NNS is sensitive and adaptable and is therefore often used as a therapeutic target to enhance early clinical outcomes, such as growth, weight gain, maturation, state control and gastric motility [9–12]. Establishing consistent and well patterned NNS is critical as NNS is a precursor to oral feeding development [13]. While intact NNS is necessary for successful oral feeding, the task of oral feeding is a more complex task and the direct associations between NNS and oral feeding skills remains mixed in the literature [14–16]. Beyond clinical

**Data Availability Statement:** The authors confirm that some access restrictions apply to the data underlying the findings. Data are confidential because the study uses personal data from minors and a clinical population. Data are available to

researchers who qualify for access to confidential data. Requests can be made by contacting: Northeastern University, 360 Huntington Ave, 525 Behrakis Building, Phone: 617-373- 4670 Email: hines.mo@husky.neu.edu.

**Funding:** EZ, Grant #: DC016030, The National Institute on Deafness and Other Communication Disorders The funders had no role in the study design, data collection and analysis, decision to publish or preparation of the manuscript.

**Competing interests:** The authors have declared that no competing interests exist

implications, NNS assessment is important as delays in NNS have been reported in approximately 35–48% of infants with different types of neonatal brain injury [17]. Thus, early NNS patterning can serve as an early marker of neonatal brain function. In addition to indicating current brain function, emerging data is available linking neonatal NNS to subsequent neurodevelopment [18]. More specifically, neonatal NNS has been associated with total motor skills, balance, total intelligence, verbal intelligence, performance intelligence, and language at age five, with better neonatal NNS relating to higher test scores [18].

It is clear that NNS is an important early clinical marker, yet there is no standardization of its measurement nor understanding of how NNS changes within a single suck sample. Furthermore, there is poor understanding of what is typical NNS beyond the neonatal period, particularly during a period of time where homeostasis has been established postpartum and the infant is becoming more self-regulated [19]. Therefore, the goal of this study was to examine how infants' NNS changes throughout a suck sample at 3-months of age. We hypothesized that as burst number increases, there would be structural changes to the NNS in cycles per burst, amplitudes, and intra-burst frequencies and that these changes would results in a decline in NNS activity. While no prior studies have examined structural changes in NNS throughout a suck sample, prior work in the oral feeding literature suggests that nutritive sucking rate declines throughout a feed in full-term [20–22] and preterm infants [23].

## Method

### Participants

Participants in this study were taken from a larger study of preterm and full-term infants that examined the relation between early sucking, oral feeding, and vocal development. Inclusionary criteria for this study included full-term (≥37 weeks' GA) infants that were 3 months (± two weeks) of age who had a least one NNS burst. Exclusionary criteria included infants born with chromosomal or congenital anomalies.

### Study design

This prospective cohort study was approved by the Institutional Review Board at Northeastern University (protocol number: 17-08-19) and parents consented for their infants to participate. Infants and their parents were recruited by word of mouth, Facebook groups, and flyer distribution. Participants' caregivers were compensated with an Amazon gift card for their time.

On the day of the study visit, the research team arrived at the infant's house approximately one hour before their scheduled feed with the custom NNS device secured in a Pelican travel case. This user-friendly device included a 0-3-month Soothie pacifier (Philips, Avent) attached, via a handle, to a pressure transducer, see pacifier image in Fig 1. The pressure transducer was housed in a black box container, which was attached to a data acquisition system (Power Lab, ADInstruments, gray box in Fig 1) that allowed for real-time visualization of the infants' suck physiology via the LabChart software (ADInstruments). This device has been approved by the biomedical team at our institution and is not commercially available nor FDA approved. Calibration was completed before every session. To calibrate, a range of pressure measurements from the system were recorded simultaneously from both the internal, uncalibrated pressure transducer, as well as an external, highly accurate and precise, calibrated pressure calibrator. This information was then used to produce a linear calibration curve for the NNS system and these values were then updated in the ADInstruments software. Once the device was set-up and calibrated, the researchers instructed the caregivers to hold the infant in a cradled position with one hand and offer him/her the research pacifier with the other hand. This position allowed for consistency of positioning across participants. Infants were then offered the

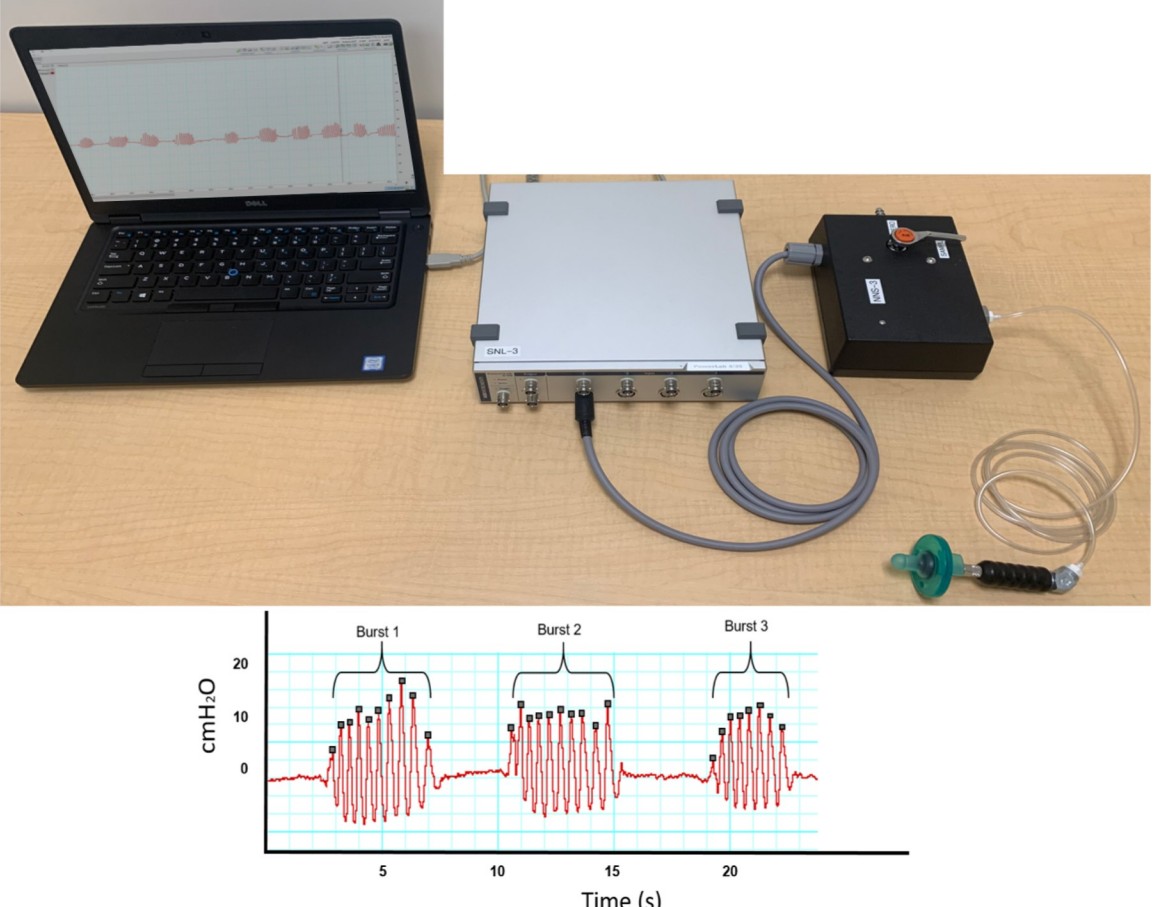

**Fig 1. NNS testing set-up.** The Soothie pacifier is attached to the pressure transducer (black box) that is connected to the data acquisition system (gray box) to allow for real time visualization of infant NNS (bottom NNS trace).

pacifier for approximately five minutes and were ideally in a quiet-alert state. Data collection was discontinued before five minutes if the infant began to cry or appeared distressed. After attaining the NNS sample, infants were offered the breast or bottle by their caregiver.

After the visit was concluded, all five minutes of the data were analyzed using LabChart software in the lab. Researchers were trained in the lab by the lab director on how to identify NNS burst in these data manually using the following criteria: bursts must contain two or more suck cycles and a cycle is considered a new burst if there is a more than an 1,000 milliseconds break between cycles. These criteria are similar to previous studies examining NNS in young infants [4, 7, 24]. Once NNS bursts were manually selected for each suck sample, they were entered into a custom NNS Burst Macro, which exported the NNS bursts data and generated the number of cycles/burst, amplitude (defined as peak height, peak-trough), and frequency for each burst. All data were saved as the participant's ID number in an effort to avoid researcher bias during data analysis.

## Sample size and statistical analyses

While no prior work has examined NNS across a suck sample, similar work has examined how NNS changes in relation to various pacifiers within 3 samples collected in six minutes using like methods. Sample size was completed from a study by Zimmerman & Barlow [25] who

examined the effect of pacifier stiffness on NNS dynamics in twenty preterm infant neonatal intensive care graduates and found a range of effect sizes (range .96–9.49) for the same five NNS dependent suck variables [25]. We conservatively estimated power of > 80% with a sample size of $N$ = 54 using the lowest effect size of (.96) found in the Zimmerman and Barlow study.

All infants who met the inclusion criteria (see previous section) were included for analysis. Statistical analysis was completed using the software package R, version 3.4.4. The maximum number of bursts recorded from a single infant was 40; however, not all infants had all 40 burst measurements due to individual differences across participants. As a result, each time point measurement was averaged across all observations and divided by the total number of individuals with that time point measurement. For example, if 48 infants had a burst measurement at burst number four, then the frequency, amplitude and cycles/burst were summed (independently) and divided by four giving an average for each outcome measure. This process was repeated for each burst number from one to 40 and these averages were then used to determine structural breaks using the software package "struccchange" [26, 27]. This software package was used to identify specific time points over the course of an infant's suck where the sucking pattern unexpectedly changed. These breakpoints were then used to specify knots in a linear regression model allowing the regression slope to change freely at the breakpoints specified.

Since the number of observed outcomes of interest declined as the number of bursts increased, we then sampled 54 individuals (with replacement) 1,000 times from the original dataset and constructed regression splines for each outcome variable within each sample. This sampling technique (bootstrapping) creates a larger dataset, allows for the construction of more reliable confidence bands, and permits inference to the entire dataset given the constraints of the study size. Knots (or, as previously described, breakpoints) were specified using the burst numbers identified from the previous breakpoint analysis. Regression splines were constructed with the R package "splines" [27]. In essence, we generated 1,000 replications of our original dataset and calculated a simple linear regression (allowing the slope to change freely at each specified breakpoint) for each outcome measure within each dataset across burst number. Predictions for each regression spline were then generated and the resulting values were aggregated in a dataset where the 50th percentile was extracted along with the corresponding 2.5 and 97.5 percentiles for the creation of confidence bands for each outcome measure. There were 1,000 generated samples in total, each with a predicted value for every outcome measure (3), and burst number (40) (for a total of 120,000 predictions).

## Results

The study consisted of 54 full-term infants (57% male, 43% female) who were seen, on average, at 92.3 days of life, or 3.03 months, (see Table 1). At time of birth, the average weight of the

Table 1. Characteristics of participants.

|  | Participants |
|---|---|
| N | 54 |
| Male / Female, Number (%) | 31 (57%) / 23 (43%) |
| Birthweight–Ounces, Mean (SD) | 121 (16.50) |
| Birth GA- Weeks, Mean (SD) | 39.3 (1.15) |
| Age at Testing–Days, Mean (SD) | 92.3 (9.53) |
| Number of Bursts, Mean (SD) | 14.5 (9.47) |
| Cycles/Burst, range | 2.00–69.00 |
| Amplitude, range | 0.55–34.60 |
| Frequency, range | 0.69–7.81 |

study infants was 121 ounces with an average 39.30 weeks' GA. A series of independent samples t-tests failed to determine any statistically significant differences between males and females with regard to: birthweight, GA, age at testing, and recorded number of bursts (see Table 2). Similarly, a series of ANOVAs failed to achieve statistical significance amongst infants when grouping infants based on the quartiles of the age at which they were tested. Finally, when individuals were grouped based on the number of bursts observed (1–10, 11–20, 21–30, and 31–40 bursts observed), no statistically significant differences were determined with regard to the aforementioned outcomes. Given the longitudinal nature of the study, only the range for the continuous study outcomes (cycles/burst, amplitude, and frequency) were reported.

All infants completed one suck sample with multiple burst measurements per sample. Of the 54 individuals: 100% of the cohort *(n = 54)* had at least one burst measurement; 89% of the cohort *(n = 48)* had at least 4 burst measurements; 50% of the cohort *(n = 27)* had at least 13 burst measurements; 33% *(n = 20)* had at least 20 burst measurements; and 2% *(n = 1)* had 40 burst measurements (Fig 2). Inter-rater reliability for NNS burst detection across two trained researchers was completed on 12/54 (22%) of the NNS data files. Inter-rater reliability was high for NNS cycles/burst (r = .97), amplitude (r = .98) and frequency (r = .92).

The breakpoint analysis identified a structural break at burst number 18 (with corresponding 95% confidence interval (CI) spanning suck bursts 16 through 23) and 34 (with corresponding 95% CI spanning suck bursts 31 through 36) for the measurement of NNS cycles/

**Table 2. Differences among infants.**

| Sex (*n*) | | Male (*n* = 31) | Female (*n* = 23) | | *p* |
|---|---|---|---|---|---|
| Birthweight—Ounces | | 123 (17.60) | 119 (14.90) | | 0.385 |
| Birth GA- weeks | | 39.20 (1.10) | 39.3 (1.23) | | 0.601 |
| Age at Testing—Days | | 90.60 (9.09) | 94.50 (9.88) | | 0.147 |
| Number of Bursts | | 14.90 (9.00) | 13.90 (10.30) | | 0.705 |
| Cycles/Burst (range) | | 2–69 | 2–52 | | - |
| Amplitude (range) | | 0.55–34.60 | 0.96–33.50 | | - |
| Frequency (range) | | 0.69–7.81 | 0.92–3.24 | | - |
| **Age (days) at Testing (*n*)** | **<85.0 (12)** | **85.0–91.5 (15)** | **91.5–99.0 (13)** | **>99.0 (14)** | ***p*** |
| Birthweight—Ounces | 118 (13.60) | 122 (18.50) | 122 (20.20) | 123 (13.60) | .832 |
| Birth GA- weeks | 39.2 (1.22) | 39.2 (.99) | 39.4 (1.04) | 39.2 (1.42) | .968 |
| Age at Testing—Days | 80.5 (3.85) | 88.3 (2.09) | 94.2 (1.86) | 105 (5.11) | - |
| Number of Bursts | 15.2 (10.00) | 15.3 (11.50) | 13.6 (6.41) | 13.9 (9.87) | .953 |
| Cycles/Burst (range) | 2–51 | 2–52 | 2–69 | 2–48 | - |
| Amplitude (range) | 1.92–32.00 | 0.94–32.50 | 0.55–31.10 | 0.93–34.60 | - |
| Frequency (range) | 0.91–4.55 | 1.00–3.91 | 0.91–3.27 | 0.69–7.81 | - |
| **Num. Bursts Observed (*n*)** | **1–10 (23)** | **11–20 (14)** | **21–30 (14)** | **31–40 (3)** | ***p*** |
| Birthweight—Ounces | 125 (14.90) | 117 (17.60) | 120 (18.10) | 124 (16.50) | .495 |
| Birth GA- weeks | 39.3 (.974) | 39.1 (1.23) | 39.4 (1.21) | 38.7 (2.08) | .772 |
| Age at Testing—Days | 92.7 (9.68) | 94 (9.50) | 89.6 (8.54) | 93.3 (15.70) | .668 |
| Number of Bursts | 5.96 | 14.70 | 23.40 | 37.30 | - |
| Cycles/Burst (range) | 2–35 | 2–69 | 2–51 | 2–48 | - |
| Amplitude (range) | .93–31.00 | .55–34.60 | 1.15–33.50 | 1.84–29.00 | - |
| Frequency (range) | .69–7.81 | .926–3.38 | 1.32–3.39 | 1.41–2.78 | - |

Unless otherwise specified, numbers are listed as Mean (SD). Independent samples t-tests were used for comparisons of sex. ANOVAs were used for the comparison of age at testing and number of bursts observed. In both instances, statistical significance was determined at the .05 level.

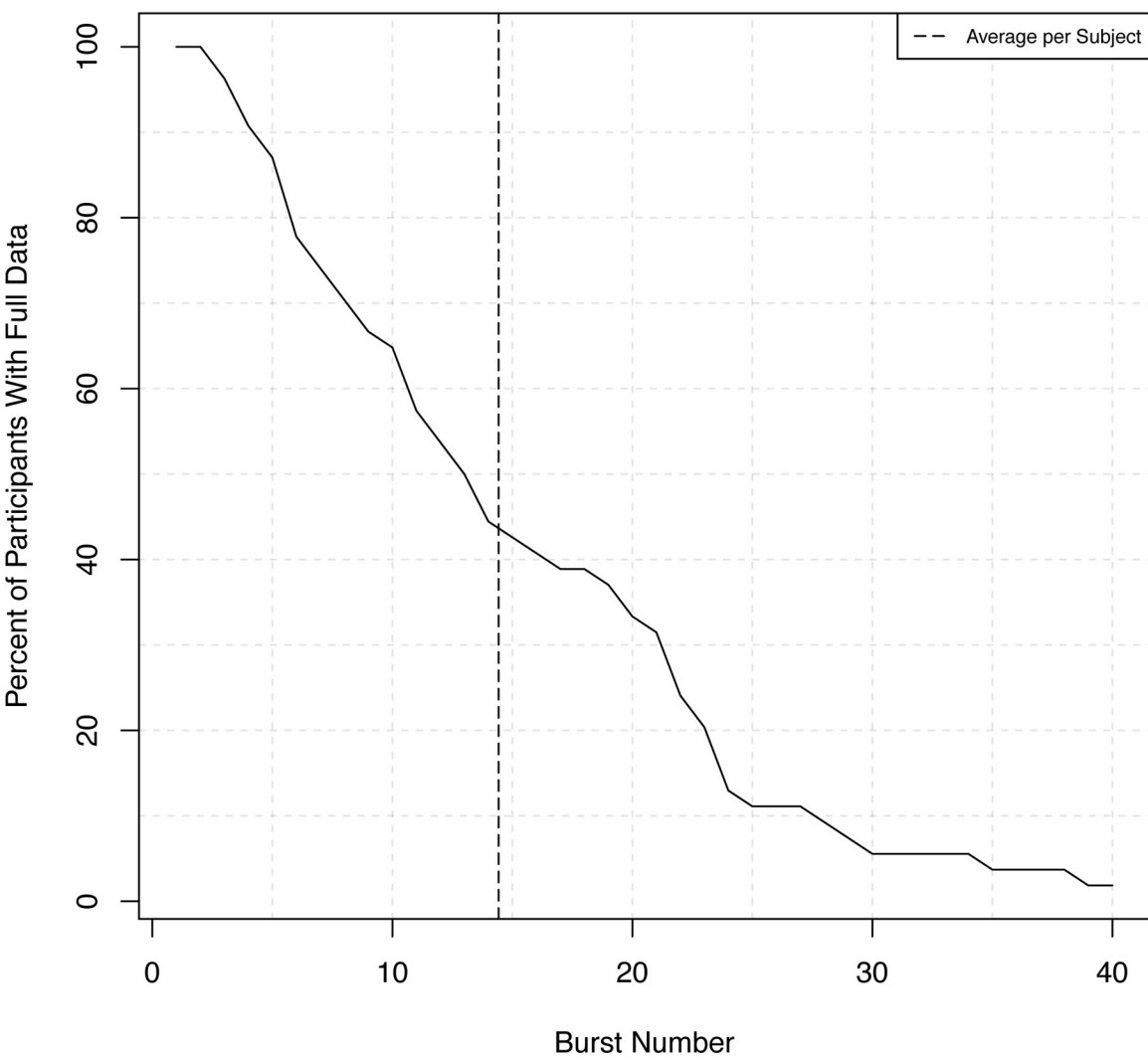

**Fig 2. Percent of participants with full data based on burst number.** The dotted line indicates the average burst number, which was 14.50.

burst. No structural breaks were determined for NNS frequency. Structural breakpoints were determined for the NNS amplitude at burst number 18 (with corresponding 95% CI spanning suck bursts 15 through 20) and burst number 29 with corresponding 95% CI between suck bursts 23 and 30.

Knots (or, specific bursts where the linear regression slope was allowed to change) were identified using the breakpoint analysis to aid in the construction of regression splines and resampling was used to generate the corresponding 95% CI for modeling the NNS cycles/burst (Fig 3) and NNS amplitude (Fig 4). No knots were determined for the NNS frequency; 95% bootstrapped CI and regression line were plotted (Fig 5).

A sensitivity analysis was completed to assess the generalizability of our results given the variability in observed number of bursts per infant. Specifically, the goal of the analysis was to investigate if breakpoints differed when varying amounts of bursts were considered. To do

## NNS Cycles/Burst w/ 95% Bootstrap CI

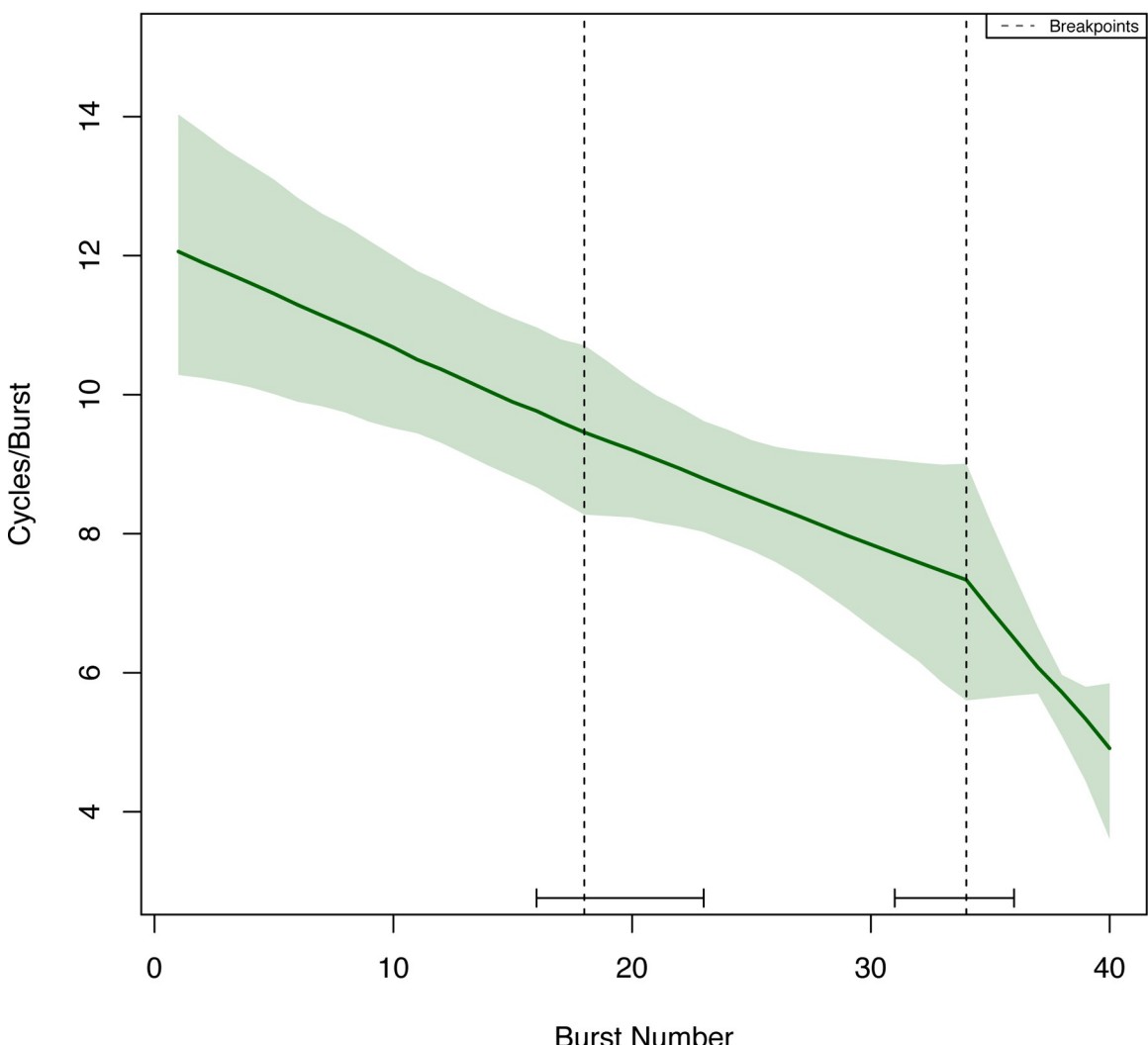

**Fig 3. Pooled NNS cycles/burst with bootstrapping.** The green shading on the pooled graphic indicates confidence intervals on each of the spline regressions found using the bootstrapped samples (2.5 and 97.5 percentile) with the dark green line indicating the median predicted value from the bootstrapped samples.

this, we considered breakpoints when observing a different number of bursts (10, 20, 30, and 40 bursts observed) and confirmed overlapping breakpoint confidence intervals for each NNS outcome between each differing number of bursts and the entire cohort (see Table 3). When only 30 bursts were observed, a structural change was found at time point 18 (confidence interval ranging from 15 to 23) for the measurement of cycles/burst and at time point 14 (confidence interval between 8 and 16) for the measurement of amplitude. With 20 bursts observed, a structural break was found for the measurement of amplitude at time points 4 and 14 with confidence intervals of 1–7 and 11–17, respectively. No breakpoint was determined for the measurement of cycles/burst when observing 20 bursts. When considering at most 10 bursts, no breakpoints were determined for any measurement. In all instances, no breakpoints were found for the measurement of frequency.

## NNS Amplitude w/ 95% Bootstrap CI

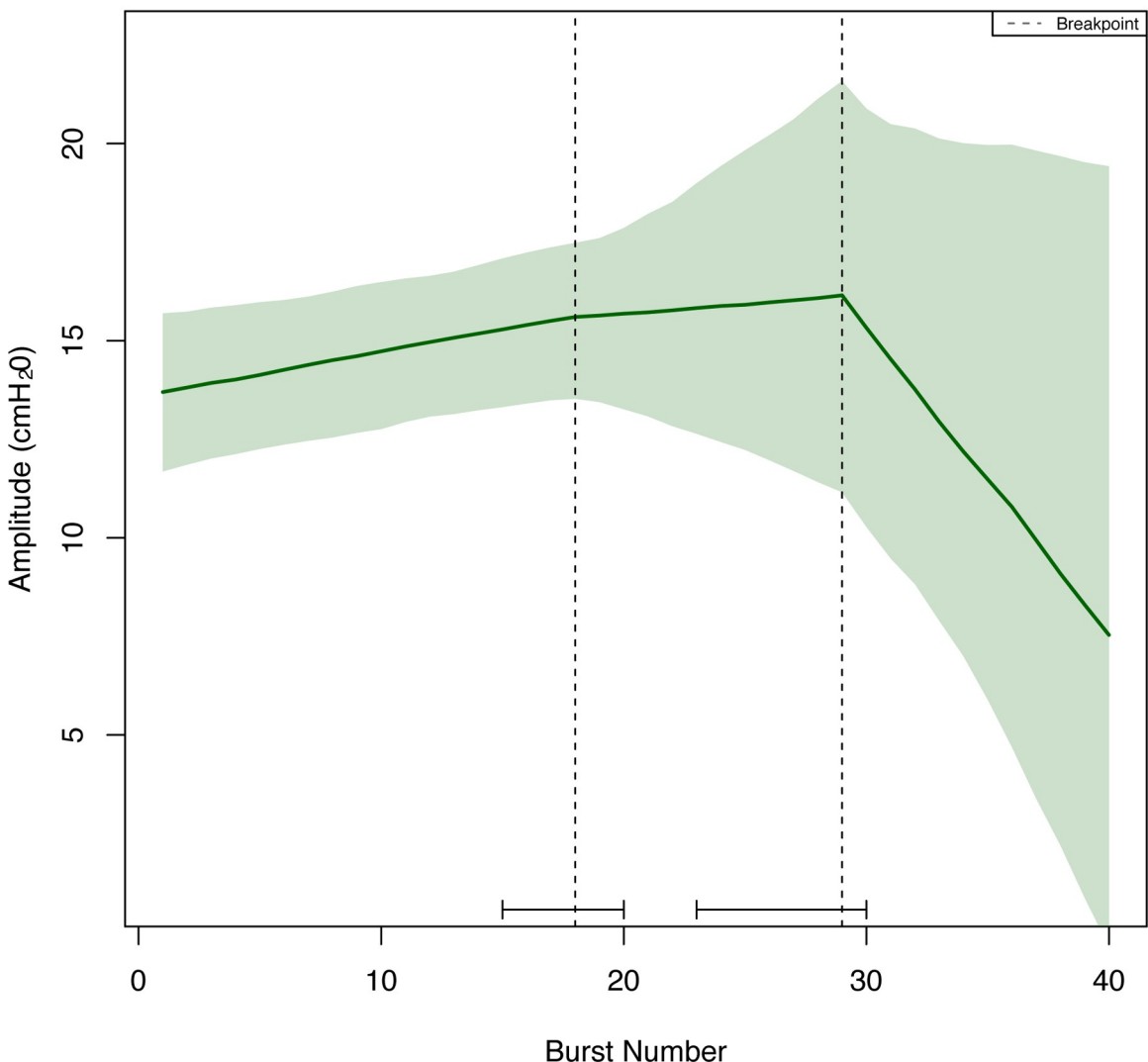

**Fig 4. Pooled NNS amplitude with bootstrapping.** The green shading on the pooled graphic indicates confidence intervals on each of the spline regressions found using the bootstrapped samples (2.5 and 97.5 percentile) with the dark green line indicating the median predicted value from the bootstrapped samples.

## Discussion

This study examined how infants' NNS changed throughout a sample in a cohort of full-term infants at 3-months. Overall, these data showed that infants exhibit changes in their NNS physiology across burst number. A first step in analyzing these data was to determine the sample size across burst number. On average, infants produced 14.50 bursts (range 1–40) during the suck sample. As burst number increased, the number of participants producing bursts decreased.

To our knowledge, this is the first time a breakpoint analysis has been used to examine infant NNS data. This type of analysis allows researchers to determine if there are instances throughout a NNS sample where cycles/burst, amplitude, or frequency change across burst

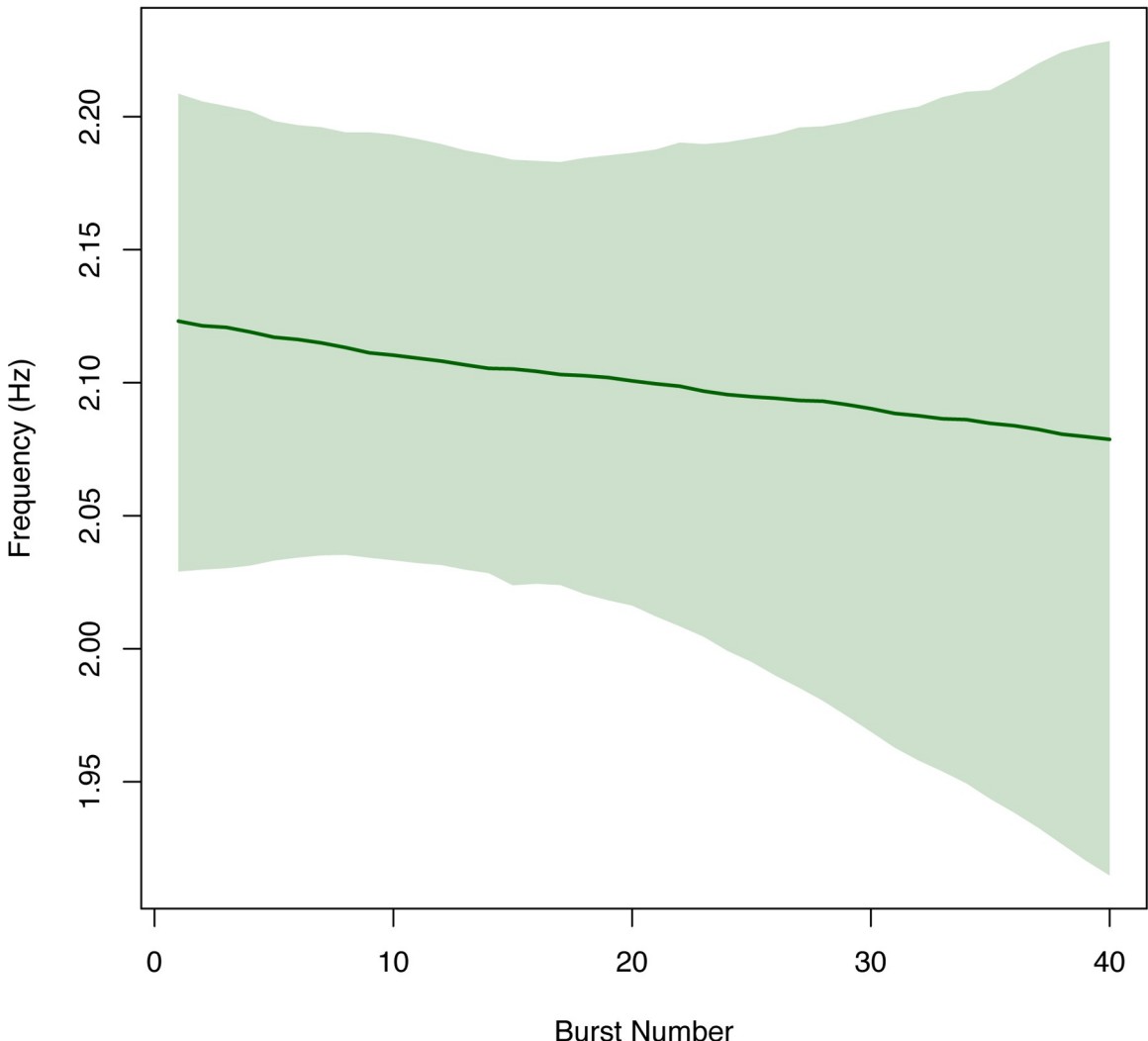

**Fig 5. Pooled NNS frequency with bootstrapping.** The green shading on the pooled graphic indicates confidence intervals on each of the spline regressions found using the bootstrapped samples (2.5 and 97.5 percentile) with the dark green indicating the median predicted value from the bootstrapped samples.

numbers to allow for a more in-depth understanding of these data structures during a sample. Results from this innovative analysis revealed breaks, or deviations, in the structure of the pooled NNS for cycles/burst and amplitude, but not for NNS frequency. These findings are somewhat in contrast to our hypotheses, which predicted that there would be structural changes to all NNS measures and that these measures would decline as bursts number increased. It appears that NNS frequency remains relatively stable as burst number increases and that these NNS variables do not all decline gradually but rather some variables increase as burst number increases and then subsequently decline (e.g., amplitude). More specifically, pooled NNS data for cycles/burst started at burst 1 with, on average, 13 cycles/burst, and this average progressively declined across burst number. The breakpoints in the data structure for this variable occurred at burst numbers 18 and 34, indicating that at these bursts there were

**Table 3. Sensitivity analysis.**

| | 40 (All) Bursts | 30 Bursts | 20 Bursts | 10 Bursts |
|---|---|---|---|---|
| **Cycles/burst** | | | | |
| Breakpoint 1 | 18 (16–23) | 18 (15–23)* | X / - | X / - |
| Breakpoint 2 | 34 (31–36) | X / - | | |
| **Amplitude** | | | | |
| Breakpoint 1 | 18 (15–20) | 14 (8–16)* | 4 (1–7) | X / - |
| Breakpoint 2 | 29 (23–30) | - | 14 (11–17)* | |
| **Frequency** | | | | |
| Breakpoint 1 | - | - | - | - |

Unless otherwise specified, numbers are listed as Breakpoint (Confidence Interval). An asterisk (*) signifies an overlapping confidence interval for breakpoints when compared to the entire (40 burst) dataset, an "X" represents a confidence interval (for a breakpoint from the entire dataset) existing beyond the scope of the data range, a dash (-) signifies a lack of breakpoint found.

significant structural changes to the cycles/burst. Pooled NNS amplitude started at burst 1 with an average amplitude of 12 cmH$_2$0. Amplitude data slowly increased until burst number 18 where a breakpoint occurred in these data. NNS amplitude continued to rise from burst numbers 18 and 28, where there was another breakpoint in these data at burst number 29 followed by a decline in amplitude from burst number 29–40. Interestingly, both NNS cycles/burst and amplitude had breakpoints in these data structures at burst number 18, indicating that there was a switch in NNS patterning behavior at this burst number (approximately halfway through the five-minute NNS sample). After burst number 18, an inverse relation occurred between NNS cycles/burst and amplitude where amplitude continued to increase and cycles/burst decreased. Thus, the increase in amplitude likely occurred as a result of the reduction in cycles/burst. This pattern was sustained until burst number 29 where amplitude quickly declined. There were no breakpoints evident for NNS frequency, which indicated no unexpected changes in the pooled NNS frequency across burst number. Previous research showed that NNS cycles/burst and amplitude were more adaptable variables, and more likely to re-organize in their data structure compared to NNS frequency, which remained relatively stable and unchanged across various sensory stimulation paradigms [4, 22].

The breakpoint analyses allowed for an in-depth view of NNS data structure across burst number; however, not all participants had the same number of bursts recorded (only 33% of the cohort had at least 20 burst measurements). Even when grouped by: sex, age at testing, and maximum number of bursts observed, our analysis showed infants did not differ with respect to: birthweight, GA, and age at time of testing. The results of the sensitivity analysis further strengthen the generalizability of the study as overlapping confidence intervals were found for measurements of both cycles/burst and amplitude when observing differing numbers of bursts. In only one instance a breakpoint was found that did not exist when analyzing the full dataset (confidence intervals of 1–7 for 20 observed bursts for the measurement of amplitude). This breakpoint suggests the existence of a possible earlier breakpoint for the measurement of amplitude; however, a larger sample size with more bursts is required to confirm the existence of this breakpoint. The sensitivity analysis did not capture a second breakpoint at timepoint 29 (with confidence interval from 23–30) for the measurement of amplitude when observing 30 bursts, though this may be because the true breakpoint exists at time point 30 (considering the confidence interval); a dataset with only 30 bursts observed would fail to include this breakpoint as it is the final timepoint in the timeseries model.

Together, the breakpoint analysis and lack of statistically significant differences amongst the study infants further strengthen the generalizability of our results despite the variable number of recorded burst measurements per infant. These analyses did not examine the range of possible values for NNS amplitude, frequency, and cycles/burst therefore bootstrapped confidence intervals for the regression splines were completed. The bootstrapping technique further allowed extrapolation to bursts with fewer observations and the repetition of constructing 1,000 regression splines provided further assurance for the plausibility of confidence bands and generalizability to the larger population. The confidence intervals revealed that NNS cycles/burst and amplitude were much less variable, with tighter confidence intervals, than NNS frequency, even though these variables had more structural changes in their data. The sets of non-overlapping confidence intervals within NNS cycles/burst and NNS amplitude identify two statistically significant time points within each outcome measurement where an actual change in sucking could be detected. While there were no large breakpoints evident in the pooled NNS frequency data structure, these raw data were more variable. This variability could be because all bursts were examined compared to other studies that measure the average per minutes or take an average over a set recording period [4, 7, 24]. When comparing our *predicted* NNS frequency data range (1.85 to 2.25 Hz) at 3-months of life to Wolff's data in 1968 elicited from full-term infants' birth to six months (2.0 to 2.8 Hz), our predicted range was smaller. This is likely due to our model, the age of participants, or the larger sample size in the present study.

The exact mechanism for why breakpoints exists for NNS across a sample remains unknown. We speculated that when breakpoints were evident, the infant was modifying their NNS for the following possible reasons: fatigue, habituation to the NNS task, state or behavioral changes, or hunger signaling. Findings from this study are similar to those in nutritive suck literature that show that infants alter their suck-swallow physiology during a feed [24–26]. It has also been shown that nutritive sucking rate declines throughout a feed in full-term [24, 25, 27] and preterm infants [23]. Further data has shown that preterm infants are more engaged during the beginning of a feed compared to the end [28]. More data is needed to explore the exact mechanism for the structural shifts across NNS burst number in these infant populations.

## Clinical implications

Results from the current study showed that burst number should be considered when assessing infant NNS. This notion is consistent with nutritive suck data that showed differences exist between judgments of swallowing physiology and the timing of fluoroscopic evaluation [29]. Furthermore, the authors state that if the fluoroscopic visualization is confined to the initial swallows of the bottle-feed, this likely limits the exam's diagnostic validity. Thus, examining a limited number of NNS bursts or starting an NNS assessment when the infant has already been sucking on a pacifier for several minutes, can result in a skewed and inaccurate representation of the infant's suck ability. This is particularly important as speech-language pathologists, nurses, and occupational therapist examine NNS coordination as part of their larger feeding assessment [30]. Therefore, it is preferable to assess the infant's suck over a period of time or a set burst range.

Additionally, these data indicated that there is an interplay between cycles/burst and amplitude at certain burst numbers. For these data, this interplay occurred after burst number 18, or approximately halfway through the suck sample. Awareness of this tradeoff is imperative for clinical practice. If a clinician is concerned about an infant's suck amplitude, but they have many cycles/bursts, they must take this into account as these data show an interaction between

these two variables across burst number. Furthermore, developmental specialists and researchers must be precise when reporting the time-frame used for analysis of the infant's NNS (e.g. the first two minutes, the middle two minutes, or the last two minutes). Lastly, it is important to consider the individual differences in all infants and their development during NNS assessment and that a one size fits all approach does not apply.

## Limitations

A number of potential limitations need to be acknowledged for this study. First, caretakers of 37 of the 54 infants in the study (68%) reported prior usage of a pacifier. For the remaining 17 infants, this could have been their first interaction with a pacifier and this could potentially alter their suck patterning. Therefore, future studies should control for previous pacifier use. In addition, this study was completed in the infant's home. While this provided them a more natural environment, there are many forms of environmental stimuli in the home that may not be controlled for and could distract the infants during the collection of the suck sample. That being said, the researchers instructed the parents on how to offer the infant the pacifier and on the level of engagement they should have during the testing. Parents offered the infant the pacifier for approximately five minutes, but this was not controlled for during the study. However, our usage of burst number rather than time allowed us to maintain consistency regardless of the sample time. Lastly, our population consisted of a homogenous demographic of infants. Therefore, these data are difficult to generalize to mothers of different ethnicities and races, marital statuses, or education levels.

## Future directions

Future studies should focus on a larger sample size, with multiple data points over time per infant to examine whether these breakpoints persist across suck samples as the infant matures and across sexes. It is imperative to determine whether different infant populations, such as infants born prematurely, have the same breakpoints in their suck data. The development of evidence-based procedural guidelines for infant NNS data collection and analyses must be established to allow for consistency across care providers.

## Conclusions

Infants' NNS changes throughout a suck sample. Infants produced, on average, 14.50 bursts during their suck sample, which lasted approximately five minutes. NNS cycles/burst and NNS amplitude had structural changes in their data across burst number, whereas NNS frequency remains relatively stable across burst number. When assessing suck, developmental specialists must observe more than just one suck burst in order to attain a more accurate view of the infant's suck patterning.

## Acknowledgments

The research team would like to thank the parents and their infants for participating in this study. We would also like to thank members of the lab for assisting with data collection and analyses.

## Author Contributions

**Conceptualization:** Emily Zimmerman.

**Data curation:** Alaina Martens.

**Formal analysis:** Emily Zimmerman, Thomas Carpenito.

**Funding acquisition:** Emily Zimmerman.

**Investigation:** Emily Zimmerman.

**Methodology:** Emily Zimmerman, Thomas Carpenito.

**Resources:** Emily Zimmerman.

**Supervision:** Emily Zimmerman.

**Validation:** Emily Zimmerman, Thomas Carpenito.

**Writing – original draft:** Emily Zimmerman.

**Writing – review & editing:** Emily Zimmerman, Thomas Carpenito, Alaina Martens.

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
