## [Decision Letter · Decision Letter 0]

3 Feb 2020

PONE-D-19-27427

Changes in Infant Non-Nutritive Sucking throughout a Suck Sample

PLOS ONE

Dear Assistant Professor Zimmerman,

Thank you for submitting your manuscript to PLOS ONE. After careful consideration, we feel that it has merit but does not fully meet PLOS ONE’s publication criteria as it currently stands. Therefore, we invite you to submit a revised version of the manuscript that addresses the points raised during the review process.

We would appreciate receiving your revised manuscript by Mar 19 2020 11:59PM. To enhance the reproducibility of your results, we recommend that if applicable you deposit your laboratory protocols in protocols.io, where a protocol can be assigned its own identifier (DOI) such that it can be cited independently in the future. For instructions see: http://journals.plos.org/plosone/s/submission-guidelines#loc-laboratory-protocols

We look forward to receiving your revised manuscript.

Kind regards,

Christos Papadelis, Ph.D.

Academic Editor

PLOS ONE

Journal Requirements:

Reviewers' comments:

Reviewer's Responses to Questions

**Comments to the Author**

1. Is the manuscript technically sound, and do the data support the conclusions?

Reviewer #1: Yes

Reviewer #2: No

2. Has the statistical analysis been performed appropriately and rigorously? 

Reviewer #1: No

Reviewer #2: Yes

3. Have the authors made all data underlying the findings in their manuscript fully available?

Reviewer #1: No

Reviewer #2: No

4. Is the manuscript presented in an intelligible fashion and written in standard English?

Reviewer #1: Yes

Reviewer #2: No

5. Review Comments to the Author

Reviewer #1: In this study, the authors investigate NNS in 3-month old infants using an instrumented pacifier and breakpoint analysis in order to explore how infants’ NNS changes throughout a suck sample.

The study is well-written; yet, the overall impact of the study is weak and doubtful. The main findings reported in the study are the breakpoints for cycles/bursts and amplitude; yet, these breakpoints are seen at burst numbers that could be recorded in a small portion of the cohort. Further analyses (such trend analysis or analysis on subset of infants with homogeneous data) must be performed to strengthen the result and the discussion.

Specific comments:

- Please explain why “Not all participants had a recorded outcome measure (NNS cycles/burst, amplitude, and frequency) across each burst”. Do the authors mean that not all participants had the same number of recorded bursts? Please rephrase or explain.

- In the methods, the authors state that “NNS Burst Macro further analyzed the NNS bursts and generated cycles/burst, amplitude (defined as peak height, peak trough), and frequency data for each sample”. If NNS cycles/burst, amplitude and frequency are computed via a software, it is unclear how the inter-rater reliability was estimated and why.

“Inter-rater reliability was high for NNS cycles/burst (r=.97), amplitude (r=.98) and frequency (r=.92).”

- The authors should explain the meaning of “knots”.

- Less than 10% of the participants had more than 29 bursts where the second breakpoint was found. Thus, the trend that the authors discuss where the amplitude drops cannot be generalized to the entire cohort, since it may be strongly biased. The same observation holds also for the first breakpoint since only 20 infants had at least 20 bursts. Different analysis that considers also time rather than only burst number should be also performed. Breakpoints should be also evaluated in terms of time.

- Authors should include a table reporting the results (outcome measure, number of bursts, duration of data) separated by main groups (e.g. by sex, age, first time users, and so on..). Further statistics on the outcome measures should be performed, to report any possible difference between groups (first time users vs. not; male vs female).

- The hypothesis reported in the introduction is not confirmed by the results. The authors should discuss this. In addition, their hypothesis regards the trend of the outcome measures (increase or decrease), while the analysis performed was focused on identifying breakpoints in a trend rather than elucidating the trend.

Reviewer #2: A better understanding of the development of NNS as well as how NNS changes within a single NNS suck sample, has strong potential for informing our knowledge of acceptable intra- and inter variability during pacifier sucking. However, the manuscript in its current form is not considered acceptable for publication. It requires major revisions.

Overall-

The study focuses on changes in NNS performance throughout a suck sample. However, in multiple places, both in the abstract and throughout the paper, the authors refer to ‘sucking’. It is recommended that they revise the manuscript throughout to be specific in noting nonnutritive sucking. A significant case in point is the suggestion that ‘infant suck can serve as a marker of neonatal brain function’. Are the authors referring to the association between nutritive sucking (NS) as a biomarker for NBI or NNS as a biomarker? This is not clear and is misleading to the reader so clarification is necessary. The manuscript would be significantly enhanced if the authors focused on the need for stable, skilled NNS as a precursor to oral feeding and use the introduction & background to establish the significance of their study within the context of that literature rather than moving back and forth from research focused on NNS and then NS.

Of additional concern is the age of many of the references used to support this work. Only the publications of these authors are current. There is a large body of more recent literature which could be used to establish the necessary background for the current study and the authors are encouraged to include those findings here. Such work would surely be useful in construction of the discussion section to address how this work supports and or refutes previous findings.

INTRODUCTION

The goal of the study is overstated. As currently written, the reader assumes what follows will be a detailed description of the change in NNS within the first year of life. Since this study only focuses at one time point- that should be clearly stated in the introduction and should also be included as part of the title.

The introduction would be enhanced if the authors provided a justification for the hypothesis.

METHODS

Necessary details regarding participants are absent. Detail is needed so readers have a better understanding of the population that was under study. ‘No known congenital anomalies’ is a very narrow inclusion criteria if the authors are hoping to inform readers regarding the NNS of health full term infants. How did you define full term? What was the mean and range of GA for the study population? Were they appropriate weight for GA? Did you exclude infants with known perinatal exposure to toxic substances, chromosomal abnormalities, etc.

The authors should provide a justification for the idea to collect this data at 3 months of age. What preliminary work has been completed that would lead you to believe that this is a critical point in the development of NNS sucking skill? Given that research suggests that nutritive suck comes under volitional control at around 4 months, it would seem that assessing NNS at that time point might yield information more suited to a discussion of potential implications for the impact on nutritive sucking.

STUDY DESIGN

Please describe the study design (prospective etc.).

Greater description is needed for all of the components involved in instrumentation (the device, data acquisition set up, the software). A diagram of the components would provide the reader with a clearer view of what the instrumentation set up looked like. For replication purposes, the following information would also be imperative: Is it commercially available? FDA cleared? What research exists to support it validity as an instrument to study NNS?

Why was the cradled position selected as the position for collecting this data? Was it convenience or the typical position in which the infant was held for feeding? Given the authors’ assertion that sensory feedback is critical to NNS, there should be some justification, as well as discussion, on the possible influence of this position on the results.

What are the references for defining the NNS criteria for analysis as stated? Describe the training that researchers went through to learn to identify NNS sucking bursts- this informs the integrity of your inter-rater reliability results.

RESULTS AND STATISTICAL ANALYSES

This section should be relabeled omitting the term RESULTS, since this is included as a separate section.

No detail on sample size calculation is provided. Please detail the appropriate parameters (randomized vs nonrandomized, power, one-tailed vs two-tailed hypothesis, significance level, estimated effect size).

Explain why all participants did not have a recorded outcome measure.

Which statistical package was used? Authors simply state ‘stats’ package.

RESULTS

The Discussion section of the paper focuses to a large extent on the clinical implications derived from this work. Yet the Statistical analysis and results sections are heavily permeated in jargon. If the audience for this work is practitioners, rather than statisticians, the authors should significantly revise these sections to provide a layman’s understanding of processes such as ‘weighted averages’, bootstrapping’, ‘constructive regression splines’, etc. Only by doing this can the average reader feel confident in the results and the ensuing interpretation.

DISCUSSION

It is particularly important in this section that the authors distinguish this work as NNS and not NS. In other words, are the authors suggesting that NNS burst number should be considered when assessing NNS suck or NS suck? Citations are necessary for the statement that ‘neurologists and other developmental specialists utilize NNS as an early tool for assessing brain injury’.

The authors make the assumption that NNS is always best at the beginning- and do not take into account the possible inter-individual differences between infants - as the NNS of some infants might be best at the midpoint in a sample or even toward the end- depending on infant state regulation at the time of data collection. Since no formal assessment of state was completed prior to data collection, this possibility should be considered in the discussion section.

Pleas revise the discussion section to separate out clinical implications, limitations and future directions.

6. PLOS authors have the option to publish the peer review history of their article (what does this mean?). If published, this will include your full peer review and any attached files.

Reviewer #1: No

Reviewer #2: No

---

## [Author Response · Author response to Decision Letter 0]

30 Mar 2020

Thank you to the associate editor and the reviewers for the opportunity to improve this manuscript. Please see the below responses to the reviewers’ comments. We feel that your insights greatly improved our manuscript.

Reviewer #1: In this study, the authors investigate NNS in 3-month old infants using an instrumented pacifier and breakpoint analysis in order to explore how infants’ NNS changes throughout a suck sample.

The study is well-written; yet, the overall impact of the study is weak and doubtful. The main findings reported in the study are the breakpoints for cycles/bursts and amplitude; yet, these breakpoints are seen at burst numbers that could be recorded in a small portion of the cohort. Further analyses (such trend analysis or analysis on subset of infants with homogeneous data) must be performed to strengthen the result and the discussion.

Response: Response: Thank you, a sensitivity analysis has been added to strengthen the generalizability of our findings.

Specific comments:

- Please explain why “Not all participants had a recorded outcome measure (NNS cycles/burst, amplitude, and frequency) across each burst”. Do the authors mean that not all participants had the same number of recorded bursts? Please rephrase or explain. Response: Thank you this has been clarified and rephrased in the manuscript.

- In the methods, the authors state that “NNS Burst Macro further analyzed the NNS bursts and generated cycles/burst, amplitude (defined as peak height, peak trough), and frequency data for each sample”. If NNS cycles/burst, amplitude and frequency are computed via a software, it is unclear how the inter-rater reliability was estimated and why.

Response: We agree that this was not clear in the text. Trained researchers picked the bursts and then the Burst macro analyzed the chosen burst and generated the other outcomes, like cycles per burst, amplitude, frequency and cycle amount. The inter-rater reliability was completed to determine that the correct burst was chosen and this would influence the subsequent measures. 

“Inter-rater reliability was high for NNS cycles/burst (r=.97), amplitude (r=.98) and frequency (r=.92).”

Response: These were completed across two trained researchers and their choosing of the bursts, see above.

- The authors should explain the meaning of “knots”.

Response: Added further clarification in text

- Less than 10% of the participants had more than 29 bursts where the second breakpoint was found. Thus, the trend that the authors discuss where the amplitude drops cannot be generalized to the entire cohort, since it may be strongly biased. The same observation holds also for the first breakpoint since only 20 infants had at least 20 bursts. Different analysis that considers also time rather than only burst number should be also performed. Breakpoints should be also evaluated in terms of time.

Response: A sensitivity analysis was added to the results section. 

- Authors should include a table reporting the results (outcome measure, number of bursts, duration of data) separated by main groups (e.g. by sex, age, first time users, and so on..). Further statistics on the outcome measures should be performed, to report any possible difference between groups (first time users vs. not; male vs female).

Response: Table 1 has been added

- The hypothesis reported in the introduction is not confirmed by the results. The authors should discuss this. In addition, their hypothesis regards the trend of the outcome measures (increase or decrease), while the analysis performed was focused on identifying breakpoints in a trend rather than elucidating the trend.

Response: Thank you this section has been edited in both the introduction and discussion sections. 

Reviewer #2: A better understanding of the development of NNS as well as how NNS changes within a single NNS suck sample, has strong potential for informing our knowledge of acceptable intra- and inter variability during pacifier sucking. However, the manuscript in its current form is not considered acceptable for publication. It requires major revisions.

Overall-

The study focuses on changes in NNS performance throughout a suck sample. However, in multiple places, both in the abstract and throughout the paper, the authors refer to ‘sucking’. It is recommended that they revise the manuscript throughout to be specific in noting nonnutritive sucking. A significant case in point is the suggestion that ‘infant suck can serve as a marker of neonatal brain function’. Are the authors referring to the association between nutritive sucking (NS) as a biomarker for NBI or NNS as a biomarker? This is not clear and is misleading to the reader so clarification is necessary. The manuscript would be significantly enhanced if the authors focused on the need for stable, skilled NNS as a precursor to oral feeding and use the introduction & background to establish the significance of their study within the context of that literature rather than moving back and forth from research focused on NNS and then NS.

Response: Thank you for this comment. We agree that the usage of sucking was confusing for the reader and this has been replaced with NNS throughout. Discussion of oral feeding and feeding has been reduced in the introduction to increase the focus on NNS. 

Of additional concern is the age of many of the references used to support this work. Only the publications of these authors are current. There is a large body of more recent literature which could be used to establish the necessary background for the current study and the authors are encouraged to include those findings here. Such work would surely be useful in construction of the discussion section to address how this work supports and or refutes previous findings.

Response: References have been updated

INTRODUCTION

The goal of the study is overstated. As currently written, the reader assumes what follows will be a detailed description of the change in NNS within the first year of life. Since this study only focuses at one time point- that should be clearly stated in the introduction and should also be included as part of the title.

Response: Thank you, this has been clarified throughout the intro and the title has been changed. . 

The introduction would be enhanced if the authors provided a justification for the hypothesis.

Response: Thank you, prior research has been added. 

METHODS

Necessary details regarding participants are absent. Detail is needed so readers have a better understanding of the population that was under study. ‘No known congenital anomalies’ is a very narrow inclusion criteria if the authors are hoping to inform readers regarding the NNS of health full term infants. How did you define full term? What was the mean and range of GA for the study population? Were they appropriate weight for GA? Did you exclude infants with known perinatal exposure to toxic substances, chromosomal abnormalities, etc.

Response: Thank you this section has been edited 

The authors should provide a justification for the idea to collect this data at 3 months of age. What preliminary work has been completed that would lead you to believe that this is a critical point in the development of NNS sucking skill? Given that research suggests that nutritive suck comes under volitional control at around 4 months, it would seem that assessing NNS at that time point might yield information more suited to a discussion of potential implications for the impact on nutritive sucking.

Response: Justification for the 3-month time point has been added to the introduction section. 

STUDY DESIGN

Please describe the study design (prospective etc.).

Response: This has been added under study design section.

Greater description is needed for all of the components involved in instrumentation (the device, data acquisition set up, the software). A diagram of the components would provide the reader with a clearer view of what the instrumentation set up looked like. For replication purposes, the following information would also be imperative: Is it commercially available? FDA cleared? What research exists to support it validity as an instrument to study NNS?

Response: More data has been added regarding the NNS devices as well as a picture, see Figure 1. 

Why was the cradled position selected as the position for collecting this data? Was it convenience or the typical position in which the infant was held for feeding? Given the authors’ assertion that sensory feedback is critical to NNS, there should be some justification, as well as discussion, on the possible influence of this position on the results.

Response: This position was chosen as it seemed the natural position for parents and infants and allowed for consistency across subjects.

What are the references for defining the NNS criteria for analysis as stated? Describe the training that researchers went through to learn to identify NNS sucking bursts- this informs the integrity of your inter-rater reliability results.

Response: References and a description of the training process have been added in the text. 

RESULTS AND STATISTICAL ANALYSES

This section should be relabeled omitting the term RESULTS, since this is included as a separate section.

Response: This section has been changed to Sample Size and Statistical Analyses

No detail on sample size calculation is provided. Please detail the appropriate parameters (randomized vs nonrandomized, power, one-tailed vs two-tailed hypothesis, significance level, estimated effect size).

Response: Details on how the sample size was determined were added. No tests of significance were included in the initial manuscript, however further clarification on confidence intervals were discussed.

Explain why all participants did not have a recorded outcome measure.

Response: If infants were crying or sleeping they did not often produce the one NNS burst during the study. These infants were excluded as they needed at least one NNS burst for inclusion into the study. This has been added to the methods.

Which statistical package was used? Authors simply state ‘stats’ package.

Response: Removed citation as this function is covered with a previous citation to the R statistical package.

RESULTS

The Discussion section of the paper focuses to a large extent on the clinical implications derived from this work. Yet the Statistical analysis and results sections are heavily permeated in jargon. If the audience for this work is practitioners, rather than statisticians, the authors should significantly revise these sections to provide a layman’s understanding of processes such as ‘weighted averages’, bootstrapping’, ‘constructive regression splines’, etc. Only by doing this can the average reader feel confident in the results and the ensuing interpretation.

Response: Jargon has been removed and a more colloquial presentation of results is presented.

DISCUSSION

It is particularly important in this section that the authors distinguish this work as NNS and not NS. In other words, are the authors suggesting that NNS burst number should be considered when assessing NNS suck or NS suck? Citations are necessary for the statement that ‘neurologists and other developmental specialists utilize NNS as an early tool for assessing brain injury’.

Response: Thank you, NNS and NS distinctions have been made clearer and a reference has been added.

The authors make the assumption that NNS is always best at the beginning- and do not take into account the possible inter-individual differences between infants - as the NNS of some infants might be best at the midpoint in a sample or even toward the end- depending on infant state regulation at the time of data collection. Since no formal assessment of state was completed prior to data collection, this possibility should be considered in the discussion section.

Response: Thank you, this section has been edited.

Pleas revise the discussion section to separate out clinical implications, limitations and future directions.

Response: Added sections in text.

---

## [Decision Letter · Decision Letter 1]

8 Jun 2020

PONE-D-19-27427R1

Changes in Infant Non-Nutritive Sucking throughout a Suck Sample at 3-Months of Age

PLOS ONE

Dear Dr. Emily Zimmerman,

Thank you for submitting your manuscript to PLOS ONE. After careful consideration, we feel that it has merit but does not fully meet PLOS ONE’s publication criteria as it currently stands. Therefore, we invite you to submit a revised version of the manuscript that addresses the points raised during the review process.

We look forward to receiving your revised manuscript.

Kind regards,

Georg M. Schmölzer

Academic Editor

PLOS ONE

Reviewers' comments:

Reviewer's Responses to Questions

**Comments to the Author**

1. If the authors have adequately addressed your comments raised in a previous round of review and you feel that this manuscript is now acceptable for publication, you may indicate that here to bypass the “Comments to the Author” section, enter your conflict of interest statement in the “Confidential to Editor” section, and submit your "Accept" recommendation.

Reviewer #1: (No Response)

2. Is the manuscript technically sound, and do the data support the conclusions?

Reviewer #1: Yes

3. Has the statistical analysis been performed appropriately and rigorously? 

Reviewer #1: Yes

4. Have the authors made all data underlying the findings in their manuscript fully available?

Reviewer #1: No

5. Is the manuscript presented in an intelligible fashion and written in standard English?

Reviewer #1: Yes

6. Review Comments to the Author

Reviewer #1: The authors addressed most of my major concerns. In particular, they added a sensitivity analysis to assess the generalizability of the results given my concern about the variability in the number of bursts per infant. The authors say that the sensitivity analysis "confirmed overlapping breakpoint confidence intervals for each NNS outcome between the subset datasets and the entire cohort". Yet, it is not clear how this analysis was performed (statistical methods) neither the quantitative results are reported.

The authors should add some more quantitative details about such analysis. The description should be added to the methods and detailed results should be reported in the text or in a table.

Regarding my previous comment: "Authors should include a table reporting the results (outcome measure, number of bursts, duration of data) separated by main groups (e.g. by sex, age, first time users, and so on..). Further statistics on the outcome measures should be performed, to report any possible difference between groups (first time users vs. not; male vs female)."

Authors added Table 1 to report main characteristics of the cohort and NNS burst number in F and M.

The other outcome measures should be also added. The authors state that no difference was found in all descriptors; however, no p-values are reported and nowhere in the manuscript they described which test was performed to test this (as mentioned also above for sensitivity analysis, statistical details should be added in methods and results)

7. PLOS authors have the option to publish the peer review history of their article (what does this mean?). If published, this will include your full peer review and any attached files.

Reviewer #1: No

---

## [Author Response · Author response to Decision Letter 1]

11 Jun 2020

hank you to the associate editor and the reviewers for the opportunity to improve this manuscript. Please see the below responses to the reviewers’ comments. We feel that your insights greatly improved our manuscript.

4. Have the authors made all data underlying the findings in their manuscript fully available?

Reviewer #1: No

Response: The authors confirm that some access restrictions apply to the data underlying the findings. Data are confidential because the study uses personal data from minors and a clinical population. Data are available to researchers who qualify for access to confidential data. Requests can be made by contacting:

Northeastern University, 360 Huntington Ave, 132

Forsyth Building, Phone: 617-373- 4670 

Email: m.hines@northeastern.edu.

Reviewer #1: The authors addressed most of my major concerns. In particular, they added a sensitivity analysis to assess the generalizability of the results given my concern about the variability in the number of bursts per infant. The authors say that the sensitivity analysis "confirmed overlapping breakpoint confidence intervals for each NNS outcome between the subset datasets and the entire cohort". Yet, it is not clear how this analysis was performed (statistical methods) neither the quantitative results are reported.

The authors should add some more quantitative details about such analysis. The description should be added to the methods and detailed results should be reported in the text or in a table.

Response: Thank you for the comment. The sensitivity analysis has been further qualified along with an addition of a table to further clarify the results (Table 3).

Regarding my previous comment: "Authors should include a table reporting the results (outcome measure, number of bursts, duration of data) separated by main groups (e.g. by sex, age, first time users, and so on..). Further statistics on the outcome measures should be performed, to report any possible difference between groups (first time users vs. not; male vs female)."

Authors added Table 1 to report main characteristics of the cohort and NNS burst number in F and M.

The other outcome measures should be also added. The authors state that no difference was found in all descriptors; however, no p-values are reported and nowhere in the manuscript they described which test was performed to test this (as mentioned also above for sensitivity analysis, statistical details should be added in methods and results)

Response: Thank you for the comment. Table 1 has been updated to include summary measures of the outcomes discussed in the submission. A second table has been included (Table 2) to further detail the lack of differences within the cohort with respect to participant sex, age, and maximum number of recorded bursts. We did not include further summary measures (beyond the range) for NNS outcome measures due to the longitudinal nature of the data. Differences between groups relating to NNS outcome measures is analyzed with the corresponding breakpoint analysis.

---

## [Decision Letter · Decision Letter 2]

23 Jun 2020

Changes in Infant Non-Nutritive Sucking throughout a Suck Sample at 3-Months of Age

PONE-D-19-27427R2

Dear Dr. Emily Zimmerman,

We’re pleased to inform you that your manuscript has been judged scientifically suitable for publication and will be formally accepted for publication once it meets all outstanding technical requirements.

Kind regards,

Georg M. Schmölzer

Academic Editor

PLOS ONE

Additional Editor Comments (optional):

Reviewers' comments:

Reviewer's Responses to Questions

**Comments to the Author**

1. If the authors have adequately addressed your comments raised in a previous round of review and you feel that this manuscript is now acceptable for publication, you may indicate that here to bypass the “Comments to the Author” section, enter your conflict of interest statement in the “Confidential to Editor” section, and submit your "Accept" recommendation.

Reviewer #1: All comments have been addressed

2. Is the manuscript technically sound, and do the data support the conclusions?

Reviewer #1: (No Response)

3. Has the statistical analysis been performed appropriately and rigorously? 

Reviewer #1: Yes

4. Have the authors made all data underlying the findings in their manuscript fully available?

Reviewer #1: Yes

5. Is the manuscript presented in an intelligible fashion and written in standard English?

Reviewer #1: (No Response)

6. Review Comments to the Author

Reviewer #1: (No Response)

7. PLOS authors have the option to publish the peer review history of their article (what does this mean?). If published, this will include your full peer review and any attached files.

Reviewer #1: No

---

## [Editor Report · Acceptance letter]

29 Jun 2020

PONE-D-19-27427R2 

Changes in Infant Non-Nutritive Sucking throughout a Suck Sample at 3-Months of Age 

Dear Dr. Zimmerman:

I'm pleased to inform you that your manuscript has been deemed suitable for publication in PLOS ONE. Congratulations! Your manuscript is now with our production department. 

Kind regards, 

on behalf of

Dr. Georg M. Schmölzer 

Academic Editor

PLOS ONE